# The transcriptome of *Listeria monocytogenes* during co-cultivation with cheese rind bacteria suggests adaptation by induction of ethanolamine and 1,2-propanediol catabolism pathway genes

Justin M. Anast [1,2], Stephan Schmitz-Esser [1,2]*

**1** Interdepartmental Microbiology Graduate Program, Iowa State University, Ames, IA, United States of America, **2** Department of Animal Science, Iowa State University, Ames, IA, United States of America

* sse@iastate.edu

**Data Availability Statement:** The raw sequencing data were deposited at the NCBI Sequence Read Archive under BioProject ID PRJNA604606.

## Abstract

The survival of *Listeria (L.) monocytogenes* in foods and food production environments (FPE) is dependent on several genes that increase tolerance to stressors; this includes competing with intrinsic bacteria. We aimed to uncover genes that are differentially expressed (DE) in *L. monocytogenes* sequence type (ST) 121 strain 6179 when co-cultured with cheese rind bacteria. *L. monocytogenes* was cultivated in broth or on plates with either a *Psychrobacter* or *Brevibacterium* isolate from cheese rinds. RNA was extracted from co-cultures in broth after two or 12 hours and from plates after 24 and 72 hours. Broth co-cultivations with *Brevibacterium* or *Psychrobacter* yielded up to 392 and 601 DE genes, while plate co-cultivations significantly affected the expression of up to 190 and 485 *L. monocytogenes* genes, respectively. Notably, the transcription of virulence genes encoding the *Listeria* adhesion protein and Listeriolysin O were induced during plate and broth co-cultivations. The expression of several systems under the control of the global stress gene regulator, $\sigma^B$, increased during co-cultivation. A cobalamin-dependent gene cluster, responsible for the catabolism of ethanolamine and 1,2-propanediol, was upregulated in both broth and plate co-cultures conditions. Finally, a small non-coding (nc)RNA, Rli47, was induced after 72 hours of co-cultivation on plates and accounted for 50–90% of the total reads mapped to *L. monocytogenes*. A recent study has shown that Rli47 may contribute to *L. monocytogenes* stress survival by slowing growth during stress conditions through the suppression of branch-chained amino acid biosynthesis. We hypothesize that Rli47 may have an impactful role in the response of *L. monocytogenes* to co-cultivation by regulating a complex network of metabolic and virulence mechanisms.

**Funding:** SSE and JMA are supported by the USDA National Institute of Food and Agriculture Hatch projects no. 1011114 and 1018898 and by the USDA National Institute of Food and Agriculture, Agricultural and Food Research Initiative Competitive Program, grant number: 2019-67017-29687.

**Competing interests:** The authors declare that they have no competing interests.

## Introduction

*L. monocytogenes* is the causative agent of the highly fatal foodborne illness listeriosis. Listeriosis affects the very young, immunocompromised, elderly, and may lead to neonatal meningitis and abortions of the fetus [1]. It is estimated that *L. monocytogenes* causes approximately 1,600 illnesses and 260 deaths in the United States each year [2] and costs an estimated $2 billion annually in health-related expenses [3]. *L. monocytogenes* is of particular concern to the food industry because of its ability to persist (i.e., the repeated isolation and survival of the same *L. monocytogenes* strain for up to several years) in food production plants [4, 5]. *L. monocytogenes* ST121 strains are among the most abundant STs isolated from the FPE [6–8] and can harbor several genes that increase their tolerance to environmental stresses associated with FPEs. Such pressures include—among others—acidic, oxidative, and disinfectant stress [9–11]. Additionally, there is the considerable challenge of competing with other resident bacteria in food and FPEs [12, 13]. Despite the importance of inter-species competition [14], research analyzing *L. monocytogenes* gene expression in co-culture with other bacteria is limited and varies in depth and methods [15–19]. Some of these studies have examined the gene expression of *L. monocytogenes* during co-cultivation with other bacteria using quantitative reverse transcriptase PCR and microarrays [15–17]. However, neither of these methods are capable of detecting novel ncRNAs. Recently, ncRNA-dependent regulation of virulence and stress tolerance responses of *L. monocytogenes* has emerged as an important topic of investigation [20, 21]. Therefore, employing methods that allow the analysis of the entire *L. monocytogenes* transcriptome should be encouraged for gene expression studies because they enable the discovery of novel transcripts. Plasmid-encoded genes of *L. monocytogenes* increase tolerance to various food production associated stresses [22, 23]. Recently, Cortes et al. 2020 [24] observed the upregulation of a putative plasmid-encoded riboswitch from an *L. monocytogenes* ST8 strain during lactic acid exposure and found that it is similar to other riboswitches involved in the response to toxic metal stress. The same authors also uncovered that the ncRNA Rli47 was by far the highest expressed chromosomal gene in an ST121 and ST8 strain [24]. Therefore, it is imperative to examine plasmid-encoded genes and ncRNAs in addition to chromosomal genes when analyzing differential gene expression patterns of *L. monocytogenes* in response to stress conditions. In this study, we sought to elucidate the differential gene expression patterns of *L. monocytogenes* strain 6179 during co-cultivation with common food bacteria through transcriptome sequencing.

## Methods

### Strains and culture conditions

The strains used in this study were *L. monocytogenes* 6179, *Psychrobacter* L7, and *Brevibacterium* S111 (Table 1). The genera *Brevibacterium* and *Psychrobacter* are found on cheese rinds and as part of the endogenous environmental taxa of cheese production facilities; thus, they may interact with *L. monocytogenes* ST121 strains in such systems [25–29] and were therefore selected for this study (see below for more details). *L. monocytogenes* 6179 is an ST121 cheese isolate from an Irish cheese production plant [30] and harbors a 62.2 kbp plasmid, pLM6179, that is involved in increased tolerance against food production-associated stress conditions [22]. The tolerance of *L. monocytogenes* 6179 towards different FPE-associated stresses has been extensively studied [10, 11, 22, 24, 31–33]. *Psychrobacter* L7 (*Gammaproteobacteria*) most likely belongs to the species *Psychrobacter celer*, and *Brevibacterium* S111 (*Actinobacteria*) may represent a novel, yet undescribed, *Brevibacterium* species. *Psychrobacter* L7 and *Brevibacterium* S111 were isolated from the rinds of Vorarlberger Bergkäse, an Austrian

**Table 1. General genetic features of the *L. monocytogenes* 6179, *Psychrobacter* L7, and *Brevibacterium* S111 genomes.**

| Strain | Assembly size in Mbp | No. of contigs | No. of predicted genes | No. of predicted plasmids | Predicted plasmid contig size(s) in kbp | Reference |
|---|---|---|---|---|---|---|
| *L. monocytogenes* 6179 | 3.01 | 2 | 3,012 | 1 | 62.2 | [36] |
| *Psychrobacter* L7 | 2.98 | 20 | 2,513 | 3 | 3.2, 3.6, 4.3, 12.5* | [35] |
| *Brevibacterium* S111 | 4.04 | 70 | 3,619 | 0 | | [34] |

*One of the three putative *Psychrobacter* L7 plasmids is separated into two contigs, therefore there are four *Psychrobacter* L7plasmid contig sizes listed.

mountain cheese produced in western Austria [34, 35]. Quantitative PCR and whole genome analysis revealed that *Psychrobacter* L7 and *Brevibacterium* S111 are highly abundant on cheese rinds throughout ripening and may significantly contribute to the cheese ripening process [34, 35].

## Co-cultivation experimental design

As only limited data on *L. monocytogenes* gene expression during co-cultivation with other bacteria is available, we designed an experimental approach that included diverse conditions to allow broad insights into the transcriptome of *L. monocytogenes*. We thus performed co-cultivations in broth (for planktonic growth conditions characterized by more transient interactions) and on agar plates to analyze growth on surfaces. *L. monocytogenes* 6179, *Psychrobacter* L7, and *Brevibacterium* S111 each grow well in Brain-Heart Infusion (BHI Thermo Scientific Remel) media. -80˚C stock cultures of *Psychrobacter* L7 and *Brevibacterium* S111 were plated on marine broth–tryptic soy agar plates consisting of Marine broth (Becton Dickinson, 40.1 g/L), tryptic soy broth (Becton Dickinson, 13.3 g/L), NaCl (30.0 g/L), agar (10.0 g/L). *L. monocytogenes* 6179–80˚C stock cultures were plated on BHI agar plates. All experimental cultivations were performed in BHI broth and plates.

L. monocytogenes 6179, *Psychrobacter* L7, and *Brevibacterium* S111 were inoculated from stock plates into 5 mL of BHI and incubated at 20˚C shaking at 200 rpm for overnight growth. 20˚C was selected for incubation to simulate temperature conditions similar to those in FPE. OD measurements were performed using a SmartSpec$^{TM}$ spectrophotometer (BIO-RAD). Overnight cultures used in broth cultivations were diluted to an $OD_{600}$ of 1.5, and 500 μL of these dilutions were added into 4 mL of BHI and incubated for two and 12 hours (h) as described in Table 2. Overnight cultures for plate cultivations were diluted to an $OD_{600}$ of 0.05 (*L. monocytogenes* 6179) or 0.5 (*Psychrobacter* L7 and *Brevibacterium* S111). 100 μL of the diluted overnight cultures used in plate co-cultivations were spread on BHI plates and cultivated for 24 and 72 h (Table 2). A monoculture of *L. monocytogenes* 6179 was grown in broth or on plates as a control. All monoculture and co-cultivations were incubated at 20˚C, and broth cultivations were shaken continuously at 200 rpm. Each experimental condition was performed in two biologically independent replicates (resulting in 24 total samples).

## RNA extraction, transcriptome sequencing, and analysis

RNA used in transcriptome sequencing was obtained using the Invitrogen Purelink RNA Mini kit. After cultivation, broth tube samples were centrifuged at 4696 x g for 3 minutes at 20˚C and pellets were immediately resuspended in 600 μL of Invitrogen Purelink RNA Mini Kit lysis buffer. For plate cultivations, a 50 μL loop of cells was collected from plates and added directly into the lysis buffer indicated above. RNA was extracted according to the manufacturer's specifications with both a chemical and mechanical lysis step using a bead-beater (Lysing

**Table 2. Cultivation conditions of *L. monocytogenes* and cheese rind bacteria with inoculation parameters reported as volume and optical density (OD$_{600}$).**

| Broth culture conditions (Two and 12 h incubations) | *L. monocytogenes* 6179 inoculum parameters | *Psychrobacter* L7 inoculum parameters | *Brevibacterium* S111 inoculum parameters |
|---|---|---|---|
| *L. monocytogenes* 6179 monoculture control | 500 µL, 1.5 OD$_{600}$ | | |
| *L. monocytogenes* 6179 and *Psychrobacter* L7 co-cultivation | 500 µL, 1.5 OD$_{600}$ | 500 µL, 1.5 OD$_{600}$ | |
| *L. monocytogenes* 6179 and *Brevibacterium* S111 co-cultivation | 500 µL, 1.5 OD$_{600}$ | | 500 µL, 1.5 OD$_{600}$ |
| Plate culture conditions (24 and 72 h incubations) | *L. monocytogenes* 6179 inoculum parameters | *Psychrobacter* L7 inoculum parameters | *Brevibacterium* S111 inoculum parameters |
| *L. monocytogenes* 6179 monoculture control | 100 µL, 0.05 OD$_{600}$ | | |
| *L. monocytogenes* 6179 and *Psychrobacter* L7 co-cultivation | 100 µL, 0.05 OD$_{600}$ | 100 µL, 0.5 OD$_{600}$ | |
| *L. monocytogenes* 6179 and *Brevibacterium* S111 co-cultivation | 100 µL, 0.05 OD$_{600}$ | | 100 µL, 0.5 OD$_{600}$ |

500 µL of diluted overnight cultures used in broth cultivations were inoculated into 4 mL of BHI broth. 100 µL of diluted overnight cultures used in plate cultivations were spread on BHI plates.

Matrix E, MP Biomedicals; Bead Mill 24 Homogenizer, Fisher Scientific). 1 µL Superase RNase inhibitor (Invitrogen) was added to each sample of eluted RNA. DNA was digested using the Turbo DNA-Free kit (Invitrogen) following the instructions of the manufacturer. A PCR targeting the *prfA* gene of *L. monocytogenes* using the primers Lip1 (5′–GAT ACA GAA ACA TCG GTT GGC– 3′) and Lip2 (5′–GTG TAA TCT TGA TGC CAT CAG G– 3′) [37] confirmed successful removal of DNA from extracted RNA samples. PCR was conducted using the Platinum Taq DNA Polymerase system (Invitrogen) according to the manufacturer's specifications. Briefly, PCR cycle conditions were as follows: initial denaturation at 94˚C (4 min), 35 cycles of denaturation at 94˚C (30 sec), annealing at 64˚C (30 sec), elongation at 72˚C (30 sec), and a final elongation step at 72˚C (5 min). RNA concentrations were then measured using a Nanodrop 2000 (Thermo Scientific). RNA integrity was verified using an RNA 6000 Nano chip on an Agilent 2100 Bioanalyzer (Prokaryote Total RNA Nano assay). Samples were depleted of rRNAs using the Illumina/Epicentre Ribo-Zero rRNA Removal Kit for gram-positive bacteria. Library preparation was conducted using an Illumina TruSeq Stranded Total RNA library preparation kit according to the manufacturer's specifications. Single-end 75 bp read length sequencing was performed using the Illumina NextSeq (v2) sequencing platform, and demultiplexing and trimming of Illumina adaptor sequences were completed by the sequencing facility (Microsynth AG, Switzerland).

Reads were mapped to their respective genomes (accession numbers: *L. monocytogenes* 6179 chromosome HG813249, plasmid HG813250; *Psychrobacter* L7 NEXR00000000; *Brevibacterium* S111 RHFH00000000) with the Burrows-Wheeler aligner [38]. Read counts per predicted gene were calculated by ReadXplorer [39], and differential gene expression analysis was performed with the DESeq2 R package by comparing *L. monocytogenes* 6179 gene expression in monoculture to gene expression in co-culture [40]. Genes were considered DE if Q-values (p-values adjusted for normalization of library size, library composition, and the correction for multiple testing by using the Benjamini and Hochberg procedure) were lower than 0.05. Principal component analysis (PCA) was conducted using the R statistical software v3.6.1 [41] and with the packages DeSeq2 v1.24.0 [40] and ggplot2 v3.2.1 [42]. *In silico* characterization of DE genes and metabolic pathways of interest were analyzed using Pfam [43] and NCBI BLASTp. Heatmaps were generated using JColorGrid [44]. Transcripts per million (TPM) were

calculated by ReadXplorer [39] to assess the magnitude of *Psychrobacter* L7 and *Brevibacterium* S111 gene expression.

## Results and discussion

### Sequence data and analysis

RNA integrity numbers (RIN) of extracted RNA ranged from 8.4–10 (averaging 9.7), indicating that the extracted RNA was of high quality. Transcriptome sequencing of *L. monocytogenes* in mono- and co-culture conditions from broth and plates resulted in 4.07 to 12.23 million reads per sample (Tables 3 and 4). PCA performed between replicates of each co-cultivation condition, and the respective condition control consistently demonstrated that replicates of shorter co-cultivations were more variable. In contrast, replicates of longer cultivations clustered much closer together by condition (i.e., co-culture replicates clustered closely together and separate from the pure culture control replicates), indicating less variability within a condition and a stronger treatment effect (S1 and S2 Figs). Among all experiments, the number of DE genes (Fig 1) and associated log2 fold changes of co-cultivation conditions ranged from 0 to 601 and -6.96 to 8.41, respectively (S1 and S2 Tables). Because the focus of this study was analyzing the gene expression of *L. monocytogenes*, we did not include monoculture control

**Table 3. Read statistics for *L. monocytogenes* co-cultivations from sequenced duplicates.**

| | Two h broth *L. monocytogenes* 6179 and *Psychrobacter* L7 | 12 h broth *L. monocytogenes* 6179 and *Psychrobacter* L7 | 24 h plate *L. monocytogenes* 6179 and *Psychrobacter* L7 | 72 h plate *L. monocytogenes* 6179 and *Psychrobacter* L7 |
|---|---|---|---|---|
| Total number of reads | 5,987,188 | 8,920,270 | 5,989,904 | 8,468,864 |
| No. of reads mapped to *L. monocytogenes* 6179 chromosome (%) | 2,669,229 (44.0%) | 6,441,382 (67.1%) | 1,203,202 (18.3%) | 2,170,764 (25.8%) |
| No. of reads mapped to *Psychrobacter* L7 (%) | 3,218,245 (54.3%) | 2,291,940 (30.6%) | 4,662,455 (79.4%) | 6,083,496 (71.3%) |
| Chromosomal coverage *L. monocytogenes* 6179 | 65 x | 157 x | 29 x | 53 x |
| Chromosomal coverage *Psychrobacter* L7 | 81 x | 58 x | 117 x | 153 x |
| No. of reads mapped to pLM6179 | 12,568 (0.21%) | 52,920 (0.60%) | 6,890 (0.12%) | 3,204 (0.04%) |
| pLM6179 coverage | 15 x | 64 x | 8 x | 4 x |
| | Two h broth *L. monocytogenes* 6179 and *Brevibacterium* S111 | 12 h broth *L. monocytogenes* 6179 and *Brevibacterium* S111 | 24 h plate *L. monocytogenes* 6179 and *Brevibacterium* S111 | 72 h plate *L. monocytogenes* 6179 and *Brevibacterium* S111 |
| Total number of reads | 6,418,506 | 6,292,883 | 6,256,047 | 7,399,414 |
| No. of reads mapped to *L. monocytogenes* 6179 chromosome (%) | 4,946,958 (76.0%) | 5,859,392 (93.1%) | 5,904,431 (94.3%) | 2,513,233 (34.0%) |
| No. of reads mapped to *Brevibacterium* S111 (%) | 1,297,706 (14.5%) | 306,896 (4.70%) | 142,728 (2.40%) | 4,584,523 (62.0%) |
| Chromosomal coverage *L. monocytogenes* 6179 | 121 x | 143 x | 144 x | 61 x |
| Chromosomal coverage *Brevibacterium* S111 | 24 x | 6 x | 2 x | 85 x |
| No. of reads mapped to pLM6179 | 25,603 (0.40%) | 51,598 (0.82%) | 31,464 (0.50%) | 11,875 (0.16%) |
| pLM6179 coverage | 31 x | 62 x | 38 x | 14 x |

Values are reported as averages of each condition replicates.

**Table 4. Read statistics of the *L. monocytogenes* monoculture controls from sequenced duplicates.**

|  | Two h broth *L. monocytogenes* 6179 | 12 h broth *L. monocytogenes* 6179 | 24 h plate *L. monocytogenes* 6179 | 72 h plate *L. monocytogenes* 6179 |
|---|---|---|---|---|
| Total number of reads | 5,466,700 | 6,686,508 | 4,940,014 | 8,331,748 |
| No. of reads mapped to *L. monocytogenes* 6179 chromosome (%) | 5,238,060 (95.8%) | 6,471,558 (96.8%) | 4,764,709 (96.5%) | 7,780,415 (93.4%) |
| Chromosomal coverage *L. monocytogenes* 6179 | 128 x | 158 x | 116 x | 190 x |
| No. of reads mapped to pLM6179 | 26,375 (0.48%) | 73,507 (1.10%) | 30,526 (0.62%) | 73,476 (0.88%) |
| pLM6179 coverage | 32 x | 89 x | 37 x | 89 x |

Values are reported as averages of each condition replicates.

samples for *Psychrobacter* and *Brevibacterium*; thus, an analysis of the DE genes of *Brevibacterium* S111 and *Psychrobacter* L7 was outside the scope of this study. However, gene expression levels of selected *Brevibacterium* S111 and *Psychrobacter* L7 genes were quantified after co-cultivation with *L. monocytogenes* 6179.

## Gene expression results unique to each co-culture condition

**L. monocytogenes 6179 gene expression changes in broth co-cultivations with Psychrobacter L7.** Reads generated during co-cultivation of *L. monocytogenes* 6179 with *Psychrobacter* L7 after a period of two h provided chromosomal coverage of *L. monocytogenes* 6179 and *Psychrobacter* L7 averaging 65 x and 81 x (Table 3), respectively. Differential gene expression analysis of *L. monocytogenes* 6179 compared to the monoculture control revealed only four DE genes (three upregulated, one downregulated; log2 fold changes from -0.92 to 1.20). Therefore, because also the log2 fold changes were low in magnitude, these DE genes are not further discussed here.

After 12 h of co-cultivation, the average chromosomal coverage of *L. monocytogenes* 6179 increased to 157 x. However, unlike co-cultivation after two h, *Psychrobacter* L7 transcriptional coverage was nearly three-fold lower (58 x). 601 *L. monocytogenes* DE genes were identified (354 upregulated and 247 downregulated) with log2 fold changes ranging from -4.1 to 6.24. tRNAs (n = 18), some ribosomal proteins (n = 5), and elongation factor Ts expression significantly increased, suggesting a general increase in the translation of mRNAs.

The hypervariable hotspot 1 of *L. monocytogenes* 6179 and other *Listeria* genomes harbors a putative type VII secretion system (*LM6179_0335–0366*) that is genetically distinct from the type VII secretion system of *L. monocytogenes* EGDe (*lmo0056-74*). Recently, the type VII secretion system of *L. monocytogenes* EGDe was shown to be dispensable during infection and even caused a detrimental effect on *L. monocytogenes* virulence during a mouse infection model study [45]. Rychli et al. 2017 [9] hypothesized that a possible function of the type VII secretion system in *L. monocytogenes* ST121 would be to mediate competition against other strains similar to a type VII secretion system found in *Staphylococcus aureus* [46]. However, only one gene, *esxA* (*LM6179_0336*, *lmo0056*), of the 6179 type VII secretion system was significantly upregulated after 12 h of co-cultivation with *Psychrobacter* L7. Overall, the 6179 type VII secretion system genes were minimally expressed, and no additional DE genes of this locus were identified in all other co-cultivation conditions analyzed in this study. Based on these results, the type VII secretion system of *L. monocytogenes* 6179 does not appear to have a specific role during co-cultivation in these experimental conditions. However, a more pronounced change in gene expression may be observed under different conditions than those

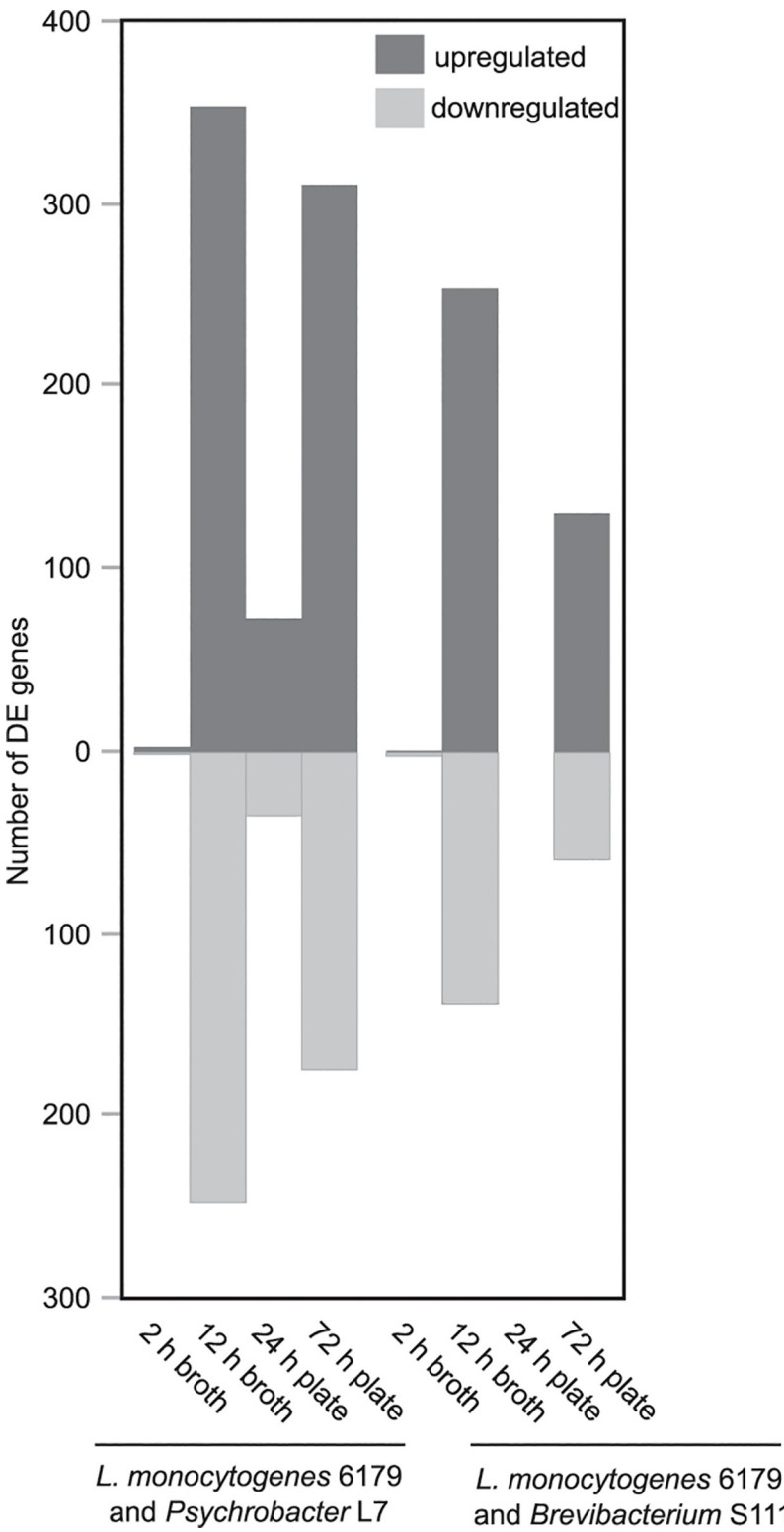

**Fig 1. The number of *L. monocytogenes* 6179 DE genes after co-cultivation with either *Psychrobacter* L7 or *Brevibacterium* S111.** Genes were considered differentially expressed if q<0.05.

applied here. Interestingly, five genes of the *L. monocytogenes* EGDe type VII secretion system were significantly upregulated in response to co-culture with *Lactobacillus casei* in the gut of mice [17]. It is also noteworthy that the *L. monocytogenes* bacteriocin Lmo2776 is absent from the genome of *L. monocytogenes* 6179. Lmo2776 was shown to reduce the growth of *Prevotella copri* and influence gastrointestinal infection [19].

**L. monocytogenes 6179 gene expression changes in broth co-cultivations with Brevibacterium S111.** Reads generated from the two h period of the broth co-cultivation of *L. monocytogenes* 6179 and *Brevibacterium* S111 provided chromosomal coverage of 121 x and 24 x for the respective organisms. Similar to the two h co-cultivation of *L. monocytogenes* 6179 and *Psychrobacter* L7, overall, only minor changes in gene expression were observed: three DE genes (two downregulated, one upregulated) with log2 fold changes ranging from -1.37 to 1.05. Due to the low number and low magnitude of log2 fold changes, these DE genes are not further discussed here.

Again, co-cultivating for a longer duration caused a greater disparity in read coverage. Co-cultivation with *Brevibacterium* S111 after 12 h revealed much higher transcriptional coverage of the *L. monocytogenes* 6179 chromosome (143 x) compared to *Brevibacterium* S111 (6 x) as well as a large increase in the number of DE genes. This analysis revealed 392 DE genes (254 upregulated, 138 downregulated) with log2 fold changes ranging from -1.83 to 2.50. Notably, tRNAs (n = 30), ribosomal proteins (n = 35), elongation factors (n = 3, Tu, G, Ts), RNA polymerase subunits (*rpoABC*), and seven genes of the *de novo* synthesis of purines pathway significantly increased in expression, suggesting a general increase in transcription and translation of *L. monocytogenes* 6179 genes. The global metabolic and virulence regulator CodY (LM6179_2018, Lmo1280) significantly increased in expression, which may suggest that in this condition, *L. monocytogenes* is starved for branched-chain amino acids. However, due to the multifaceted role of CodY in both regulation of many stress and virulence genes, we can only hypothesize a role for CodY in co-culture [47].

Recently, a 12.5 kbp insertion that harbors six genes, including a nine kbp putative rearrangement hot spot (RHS) protein (LM6179_0173) and a putative RNA 2'-phosphotransferase, was identified in the genomes of ST121 strains [9, 36]. This region was suggested to be involved in competition because several other bacterial RHS proteins mediate both inter- and intra-species competition by exhibiting nuclease activity [48, 49]. However, during the co-culture of *L. monocytogenes* 6179 with *Psychrobacter* L7 and *Brevibacterium* S111, *LM6179_0173* and the other genes in this locus were either significantly downregulated or had no significant change in expression. Furthermore, genes of the RHS loci had very low expression levels overall. These results contradict the hypothesis that the RHS locus in *L. monocytogenes* ST121 strains, at least under the conditions applied in the present study, is involved in competition with other bacteria. The function of the RHS locus needs to be clarified in future studies.

It should be noted that two h of co-cultivation with both *Psychrobacter* L7 and *Brevibacterium* S111 may have been too brief to induce significant changes in the transcriptome of *L. monocytogenes* 6179. For future analysis of *L. monocytogenes* 6179 differential gene expression in response to short-term exposure of co-cultivation conditions, a more concentrated initial inoculum of bacteria may provide better insight. In contrast, 12 h of co-cultivation induced massive gene expression shifts in the transcriptome of *L. monocytogenes* 6179.

**L. monocytogenes 6179 gene expression changes in plate co-cultivations with Psychrobacter L7.** Transcriptome sequencing of RNA extracted from the co-cultivation of *L. monocytogenes* 6179 with *Psychrobacter* L7 for a period of 24 h on plates provided chromosomal coverage of *L. monocytogenes* 6179 and *Psychrobacter* L7, averaging 29 x and 117 x, respectively (Table 3). This condition revealed 108 *L. monocytogenes* 6179 DE genes (73 upregulated, 35 downregulated) with log2 fold changes ranging from -3.27 to 8.41. In general, housekeeping

genes involved in fermentation, zinc transport, and pyrimidine synthesis were upregulated, while genes involving sulfur-containing amino acid transport and heme-degradation were among the most downregulated.

A gene encoding a putative multidrug resistance protein (*norB*, *LM6179_0238*, *lmo2818*) was significantly upregulated with a log2 fold change of 2.83. NorB shares 57% amino acid identity to the functionally characterized NorB of *Staphylococcus aureus*, which contributes to bacterial fitness in abscesses and resistance to antimicrobials [50]. Other studies demonstrated that *norB* was highly upregulated during exposure to acidic conditions and hypoxic conditions, suggesting that NorB may be involved in stress response [51, 52].

The *Psychrobacter* L7 genome encodes a putative type VI secretion system (Locus_tags: *CAP50_05950* to *CAP50_05985*, TPM: 35 to 7423) and a putative bacteriocin production protein (CAP50_06600, PFAM family: PF02674; TPM: 37 to 67) that were expressed after 24 h of co-cultivation. Type VI secretion systems have been shown to inject bactericidal toxins into the cytoplasm of competitor cells [53]. Two genes (*CAP50_05975* (TPM: 6941 to 7423); *CAP50_05980* (TPM: 959 to 1019)) of the putative type VI secretion system, were among the 10% highest expressed genes in the *Psychrobacter* L7 transcriptome after 24 h of co-cultivation on plates. The upregulation of these genes may indicate that the putative type VI secretion system is expressed in response to *L. monocytogenes*. However, a monoculture control of *Psychrobacter* L7 would be required to verify this hypothesis. Schirmer et al. 2013 [26] found that *Psychrobacter spp.* isolated from a drain in a cheese ripening cellar exhibited antilisterial properties. Antimicrobial activity of *Psychrobacter* species has been reported in additional studies; however, the genes encoding the antimicrobial compounds have yet to be identified [54, 55].

After 72 h of co-cultivation transcriptional coverage of *L. monocytogenes* 6179 increased with an average chromosomal coverage of 53 x. Similar to the shorter plate co-cultivation of 24 h, *Psychrobacter* L7 transcriptional coverage was nearly three-fold higher (153 x). Co-cultivation after 72 h resulted in 485 *L. monocytogenes* 6179 DE genes (311 upregulated, 174 downregulated) with log2 fold changes ranging from -6.96 to 6.47. tRNA (n = 17) and ribosomal protein (n = 25) gene expression significantly increased, suggesting increased translational activity under these conditions.

The genome of *L. monocytogenes* 6179 harbors three large prophages that can be highly conserved among ST121 strains. These prophages are inserted directly downstream of the tRNA Arg-TCT, Arg-CCG, and Thr-GGT genes [9, 36]. A high proportion of genes (n = 26 to 29; out of 60 to 68 total genes) from each prophage was significantly upregulated after 72 hours of plate co-cultivation with *Psychrobacter* L7 (S3 Table). The upregulation of 14 *lma* prophage genes, which encode a bacteriocin, and are also referred to as monocin locus [56, 57], might suggest a direct antagonistic response to *Psychrobacter* L7 (S1 Table) [58, 59]. The *lma* prophage has been suggested to be important in the pathogenic life-cycle of *L. monocytogenes* [59, 60]. Additionally, a previous study revealed increased expression of the *lma* prophage during acid stress exposure [61], indicating that the *lma* prophage may be important in *L. monocytogenes* beyond pathogenesis. Such results also further support the concept that some prophages can provide beneficial effects for the bacterial host during stress conditions [62, 63]. Recently, Argov et al. 2019 [57] discovered that the *lma* prophage controls the induction of the lytic *comK* prophage in *L. monocytogenes* 10403S, thus synchronizing their lysis modules. Argov et al. 2019 [57] further suggest that the co-induction of the active lytic phage and the monocin may enhance bacterial fitness under stress. Additionally, bacteriocins are known to provide a competitive advantage by killing neighboring cells. Therefore, the monocin may enhance fitness in diverse bacterial communities [57]. However, the *comK* prophage is absent from the *L. monocytogenes* 6179 genome. It is tempting to speculate that the simultaneous induction of the three *L. monocytogenes* 6179 prophages and the *lma* locus may be coordinated

in a similar mechanism suggested by Argov et al. 2019 [57] in response to *Psychrobacter* L7. Therefore, the global induction of *L. monocytogenes* 6179 prophage genes observed in this study suggests a possible benefit during exposure to stresses that co-cultivation conditions may elicit; however, further experiments are required to verify this hypothesis.

The *L. monocytogenes* ncRNA Rli47 was significantly upregulated (3.49 log2 fold change) after co-cultivation with *Psychrobacter* L7 (72 h). Remarkably, an average of 72% of all *L. monocytogenes* chromosomally mapped reads belonged to Rli47 in both 72 h monoculture and co-culture replicates. Rli47 is a *trans*-acting ncRNA 515 nucleotides in length and is located in the intergenic region between *lmo2141* and *lmo2142* [64]. The expression of Rli47 is under the control of a $\sigma^B$ regulated promoter, suggesting a role in stress response. In line with this, Rli47 is expressed during various stress conditions, including acidic [24], oxidative [65], stationary-phase [65–67], in the gastrointestinal tract [66], and during intracellular replication in macrophages [68]. Marinho et al. [64] discovered that Rli47 hinders *L. monocytogenes* growth during harsh conditions by the suppression of branched-chained amino acid biosynthesis. Additionally, in the same study, it was revealed that Rli47 influences the expression of over 150 genes that are involved in amino acid metabolism and transport, electron transport, fermentation, chorismate biosynthesis (*aro* pathway), and purine biosynthesis. Interestingly, the Rli47 regulon largely overlaps (n = 42 genes) with that of CodY, further establishing a possible role of Rli47 in the global regulation of metabolism during stress conditions. Rli47 was also upregulated during the co-culture of *L. monocytogenes* with *Lactobacillus* in the lumen of gnotobiotic mice [17], suggesting a possible link between stress response, virulence, and interaction with other bacteria. These data indicate that Rli47 has an important role in gene regulation during virulence and stress exposure, and may be involved in adapting to the complex and transient metabolite pool presented by co-culture conditions. However, the putative role of Rli47 modulating gene expression in response to co-culture will need to be verified by subsequent studies.

**L. monocytogenes gene expression changes in plate co-cultivations with Brevibacterium S111.** Plate cultivation of *L. monocytogenes* 6179 and *Brevibacterium* S111 after 24 h revealed much higher transcriptional coverage of the *L. monocytogenes* 6179 chromosome (144 x) compared to *Brevibacterium* S111 (2 x). These coverage data are in contrast with plate co-cultivations of *L. monocytogenes* 6179 and *Psychrobacter* L7 after 24 h but are similar to *Brevibacterium* S111 after 12 h in broth. No *L. monocytogenes* 6179 DE genes were identified in this co-cultivation condition. The absence of DE genes could be because *L. monocytogenes* 6179 was under minimal pressure to change its gene expression due to a very low abundance of or minimal transcriptional activity from *Brevibacterium* S111, indicated by its very low transcriptional coverage.

Interestingly, the chromosomal coverage of *Brevibacterium* S111 (85 x) surpassed *L. monocytogenes* 6179 (61 x) after 72 h and induced the DE of 190 *L. monocytogenes* genes (131 upregulated, 59 downregulated). Log2 fold changes of DE genes ranged from -2.39 to 7.83. DE genes of interest from this co-culture condition are mentioned in the shared DE genes section (see below).

Notably, a *Brevibacterium* S111 homolog (EB836_RS03155, 87% amino acid identity) of Linocin M18, a *Brevibacterium linens* anti-listerial bacteriocin, was expressed by *Brevibacterium* S111 in all but one (24 h plate co-cultivation) replicate of co-cultivation with *L. monocytogenes* 6179 in both broth and plate conditions. *EB836_RS03155* was among the 20% highest expressed genes in the 72 h replicates of *Brevibacterium* S111. The expression of *EB836_RS03155* was most pronounced after 72 h of co-cultivation with *L. monocytogenes* 6179 on plates (TPM: 152 to 191) and substantially increased compared to the 24 h time point (TPM: 0 to 23.5). Linocin M18 has been found to inhibit the growth of *L. monocytogenes* and

other *Listeria* species [69]. Therefore, the transcription of *EB836_RS03155* may suggest that *Brevibacterium* S111 is producing this antilisterial bacteriocin in response to co-culture with *L. monocytogenes* 6179. However, similar to the previously mentioned putative bacteriocin of *Psychrobacter* L7, a monoculture control of *Brevibacterium* S111 gene expression would be required to validate this hypothesis.

## Plasmid gene expression

*L. monocytogenes* plasmid reads accounted for 0.04 to 1.10 percent of the total reads per sample, and plasmid coverage ranged from 4 to 89 x (Tables 3 and 4). *L. monocytogenes* 6179 exposed to co-culture conditions induced the differential expression of seven plasmid genes (Table 5). An uncharacterized *uvrX* gene (*LM6179_RS15380*) was induced by both 12 h broth co-cultivations with log2 fold changes of 2.16 (*Psychrobacter* L7) and 1.16 (*Brevibacterium* S111). *uvrX* is described by Kuenne et al. [70] to be highly conserved in *Listeria* plasmids and may be a DNA polymerase IV and harbors three putative protein domains predicted to be involved in ultraviolet radiation protection (PFAM domains PF00817, PF11799, and PF11798). It should be noted that it is currently unknown if the plasmid-encoded *L. monocytogenes uvrX* gene is involved in stress response or plasmid maintenance. Data from previous studies show inconsistent gene expression patterns of *uvrX* during the stress response of *L. monocytogenes*. Cortes et al. [24] observed that *uvrX* was induced during lactic acid stress. However, Hingston et al. [23] found that during mild acid stress, *uvrX* expression was highly strain-dependent.

Genes of a putative toxin-antitoxin cassette (*LM6179_RS15455*, *LM6179_RS15450*) were significantly upregulated after co-cultivations with *Psychrobacter* L7 for 72 h on plates. Homologs of this toxin-antitoxin cassette are present within many *Listeria* plasmids [23, 70]. Similar toxin-antitoxin cassettes have been characterized in *E. coli* and in *Lactobacillus salivarius* plasmids to be involved in plasmid maintenance and stress response [71, 72]. In *L. monocytogenes* 6179, both genes of the putative toxin-antitoxin cassette were significantly upregulated during lactic acid stress, while the plasmid replication protein (RepA) was significantly downregulated. The downregulation of *repA* suggests a decrease in plasmid replication under stress conditions. These data indicate that the toxin-antitoxin cassette has a role in stress response and probably not plasmid maintenance [24]. Future studies are needed to examine the function of the plasmid-borne putative toxin-antitoxin system in *L. monocytogenes*.

**Table 5. Differentially expressed *L. monocytogenes* pLM6179 plasmid genes during co-cultivation with cheese rind bacteria.**

| *L. monocytogenes* pLM6179 locus_tag | Product | *L. monocytogenes* 6179 and *Psychrobacter* L7 12 h | *L. monocytogenes* 6179 and *Psychrobacter* L7 72 h | *L. monocytogenes* 6179 and *Brevibacterium* S111 12 h |
|---|---|---|---|---|
| *LM6179_RS15380* | Putative lesion bypass phage DNA polymerase (UvrX) | 2.16 | ns* | 1.16 |
| *LM6179_RS15385* | conserved protein of unknown function | 1.85 | ns* | 0.95 |
| *LM6179_RS15455* | Death on curing protein–Doc toxin | ns* | 1.83 | ns* |
| *LM6179_RS15450* | Prevent host death protein–Phd antitoxin | ns* | 1.87 | ns* |
| *LM6179_RS15345* | cadmium-transporting ATPase Tn*5422* | ns* | -1.28 | ns* |

*ns denotes that the gene was not DE in that condition.

Values show the log2 fold changes in gene expression.

Notably, although not a DE gene, reads mapped to the *clpL* gene (*LM6179_RS15400*) accounted for 30–80% of total reads mapped to pLM6179 in all replicates (including the controls). The identical *clpL* gene on plasmid pLM58 is responsible for heat tolerance, and highly similar *clpL* genes in *Lactobacillus* and *Streptococcus* are involved in general stress response [73–75]. High expression levels of *clpL* have been observed previously during oxidative stress and significant upregulation and high expression levels after lactic acid exposure [24]. The high expression levels of plasmid *clpL* genes suggest importance as constitutively expressed genes not only during stress conditions but also under non-stress conditions. Overall, the differential expression of 6179 plasmid genes during co-culture was not as striking when compared to other stress conditions (e.g., lactic acid stress [24]). However, minor expression changes of genes suspected to be involved in stress response were observed.

**Shared changes in L. monocytogenes gene expression during different co-cultivation conditions.** The co-cultivation of *L. monocytogenes* 6179 with *Psychrobacter* L7 or *Brevibacterium* S111 induced differential expression patterns of several genes and pathways conserved among multiple co-cultivation conditions, which are described below and summarized in S4 Table.

**Virulence genes.** A gene annotated as a cell wall surface protein (*LM6179_0811*) was significantly upregulated in all plate co-culture conditions, excluding 24 h with *Brevibacterium* S111. LM6179_0811 is a truncated homolog of Lmo0514, a protein that is essential during murine infection and increased survival during acidic conditions [76, 77]. Rychli et al. [9] hypothesized that the truncation of LM6179_0811, and the resultant absence of the LPXTG cell wall anchor domain that is found in Lmo0514, could contribute to the attenuated virulence of ST121 strains. Additionally, the truncated Lmo0514 homologs may confer a selective advantage outside of the infectious life-cycle of *L. monocytogenes*, which is supported by the upregulation of *LM6179_0811* during plate co-cultivations.

Listeriolysin O (LLO, *hly*, LM6179_0492) is a pore-forming toxin that is essential for the escape of *L. monocytogenes* from vesicles within host cells [1]. *Hly* was significantly upregulated in both co-cultivations with *Psychrobacter* L7 and *Brevibacterium* S111 after two h in broth and after 72 h on plates. Stress conditions unrelated to infection, including co-cultivation with *Bifidobacterium breve* are known to induce *hly* expression [16, 78, 79]. However, a function of LLO during the saprophytic life-cycle of *L. monocytogenes* has yet to be elucidated. The *Listeria* adhesion protein (LAP, LM6179_2386, Lmo1634), was significantly upregulated in all co-cultivation conditions save for co-cultivation with *Brevibacterium* S111 for two h in broth and 24 h on plates where very few or no DE genes were observed. *Lap* was the highest upregulated gene (24 h of co-cultivation with *Psychrobacter* L7, log2 fold change of 8.41) observed throughout all co-cultivation conditions in this study. LAP mediates the translocation of *L. monocytogenes* through epithelial tight junctions during infection and enables *L. monocytogenes* to systemically spread [80, 81]. LAP is also a functionally characterized alcohol-acetaldehyde dehydrogenase [80]. Additionally, co-culturing *L. monocytogenes* with *Lactobacillus* in the intestine of gnotobiotic mice induced the expression of *lap* [17]. Inferences from the currently presented data and from previous research that revealed *lap* is upregulated during nutrient limitation [82] leads us to suggest that the upregulation of *lap* in this study may be due to nutrient acquisition stress as a consequence of co-culture. Notably, *lap* was found to be the most impacted Rli47-dependent gene in the Rli47 deletion mutant study mentioned above [64], and Cortes et al. [24] highlight a putative consistent regulatory relationship of Rli47 with LAP among transcriptional studies. These data suggest that Rli47 may directly or indirectly regulate LAP—among others—under various stress conditions, including co-cultivation with other bacteria.

**Metabolism: Respiration and fermentation.** Culturing *L. monocytogenes* 6179 with *Psychrobacter* L7 and *Brevibacterium* S111 significantly altered the expression of electron transport chain systems and fermentative pathway genes. Genes encoding a cytochrome *bd*-type oxidase (*cydAB*) and the ABC transporter *cydCD*, which is essential for CydAB insertion into the membrane, were significantly upregulated in both 12 h broth co-cultivations with log2 fold changes ranging from 0.58 to 1.96. Interestingly, all four genes of the second *L. monocytogenes* cytochrome terminal oxidase (QoxABCD; cytochrome $aa_3$ menaquinol oxidase) were downregulated in co-cultivation with *Psychrobacter* L7 for 12 h in broth. However, after co-cultivation with *Psychrobacter* L7 for 24 h on plates and co-cultivation with *Brevibacterium* S111 in broth for 12 h, only *qoxA* decreased in expression. Unlike QoxABCD, CydAB is essential for aerobic and intracellular growth and increases tolerance towards reactive nitrogen species in *L. monocytogenes* [83]. The upregulation of the cytochrome bd complex in 12 h broth co-culture suggests that, in *L. monocytogenes* 6179, the production of ATP is occurring via oxidative phosphorylation with oxygen as the terminal electron acceptor.

*L. monocytogenes* can also use extracellular terminal electron acceptors such as iron and fumarate in the extracellular electron transport chain (EET). The transfer of electrons to extracellular fumarate or iron is mediated through two surface-associated proteins: FrdA (fumarate reductase: LM6179_0655, Lmo0355) and PplA (LM6179_0048, Lmo2636) [84, 85]. Electrons derived from NADH are segregated from aerobic respiration by a specialized NADH dehydrogenase (Ndh2, LM6179_0049, Lmo2638) that shuttles them to a specific quinone pool which in turn transfers the electrons to PplA and FrdA [84, 85]. Both 12 h co-cultivation conditions significantly induced the expression of *frdA* and *ndh2*; however, *pplA* only significantly increased in expression after co-cultivation with *Brevibacterium* S111 after 12 h. *frdA* and *pplA* were also upregulated after 24 h of co-cultivation with *Psychrobacter* L7. Interestingly, *pplA* was suppressed after 72 h of co-cultivation with *Psychrobacter* L7 on plates, and *ndh2* was downregulated after both 72 h conditions. These data suggest that at least in broth co-cultivations, *L. monocytogenes* 6179 utilizes both aerobic and anaerobic respiration systems.

Co-culturing with cheese rind bacteria may also have led *L. monocytogenes* 6179 to increase pyruvate fermentation. During pyruvate fermentation, pyruvate may be catabolized to formate and acetyl-CoA by pyruvate formate-lyase (PflABC, LM6179_2686, Lmo1917; LM6179_2150, Lmo1406; LM6179_2151, Lmo1407). *pflABC* were significantly upregulated after 12 h of broth co-culture with both *Psychrobacter* L7 and *Brevibacterium* S111 and after 24 h of plate co-cultivation with *Psychrobacter* L7. *L. monocytogenes* 6179 genes annotated as formate dehydrogenases (*fdhA LM6179_1612*, *lmo2586*; and *fdhD LM6179_1614*, *lmo2584*) were significantly upregulated after co-cultivation with *Psychrobacter* L7 (*fdhA* and *fdhD*) and *Brevibacterium* S111 (*fdhA* only) for 12 h in broth. Homologs of *fdhAD* are found in the functionally characterized formate dehydrogenases of *Wolinella* (*W.*) *succinogenes* and *E. coli* [86]. Anaerobic respiration with formate as an electron donor and menaquinone as the electron carrier is catalyzed by two *fdhEABCD* operons in *W. succinogenes*. In *E. coli*, FdhD is a sulfurtransferase that is required for formate dehydrogenase to be functional. Additionally, formate dehydrogenase increased tolerance to oxidative and stationary phase stress in *E. coli* [87]. The expression of a putative formate/nitrite transporter (LM6179_1228, Lmo0912, PF01226) was significantly upregulated in both 12 h co-cultivation conditions and after 24 h of co-cultivation with *Psychrobacter* L7. It is tempting to speculate that the putative formate dehydrogenase of *L. monocytogenes* 6179 may transfer electrons from formate produced by pyruvate fermentation to an unidentified quinone acceptor during energy metabolism; a hypothesis that has been previously suggested by others [88].

The gene expression of *frdA*, *pflABC*, *pplA*, and the putative formate/nitrite transporter significantly altered in a Rli47 deletion mutant study mentioned previously, suggesting that Rli47

may have a broad role in modulating energy metabolism [64]. Perhaps most notably, Rli47 influences the transcription of four key *aro* pathway genes. Mutants of the *aro* pathway are highly attenuated in mouse infection models due to the inability to synthesize chorismate, a required precursor for menaquinone production [89]. Menaquinone is essential during cellular respiration in *L. monocytogenes*; thus, regulation via Rli47 would influence quinone pools of respiration systems (both aerobic and anaerobic). During co-cultivation, the expressions of *aroAEF* significantly altered in expression, suggesting an alteration in menaquinone production, possibly mediated by Rli47.

The present study reveals that *L. monocytogenes* is altering expression of aerobic and anaerobic respiratory and pyruvate fermentation pathways in response to co-culture conditions. Changing the expression of various metabolic pathway genes may enable *L. monocytogenes* to adapt better to nutrient availability during co-culture, and it seems that Rli47 may have a general role in modulating these changes.

**Pyrimidine biosynthesis.** A gene cluster (*pyr* genes) annotated as being involved in pyrimidine biosynthesis and salvage was upregulated in both 12 h co-cultivations, 24 h plate co-cultivation with *Psychrobacter* L7, and 72 h of co-cultivation with *Brevibacterium* S111 (Table 6). We hypothesize that *L. monocytogenes* 6179 is experiencing nitrogen starvation in co-culture and therefore induces the pyrimidine utilization cluster to derive nitrogen from glutamine. Tognon et al. [90] suggested that glutamine may act as a nitrogen and carbon source during nitrogen starvation and that nitrogen starvation itself is a strong inducer of this gene cluster. A precedent for pyrimidine utilization in co-culture already exists: When *Staphylococcus* (*S.*) *aureus* was co-cultivated with *Pseudomonas aeruginosa*, the highest induction of *S. aureus* gene expression was in the pyrimidine biosynthesis and salvage pathway [90]. Finally, the hypothesis that *L. monocytogenes* may undergo nitrogen starvation during co-cultivation may be further supported by the observed induction of the ethanolamine degradation genes discussed below. Kutzner et al. [91] demonstrated that *L. monocytogenes* could replace glutamine with ethanolamine as a nitrogen source. Therefore, we argue that nitrogen derived from the pyrimidine biosynthesis and ethanolamine degradation gene clusters may confer a selective advantage for *L. monocytogenes* during nitrogen starvation as a consequence of co-culture.

**Cobalamin-dependent gene cluster.** Perhaps the most notable shift in gene expression during co-cultivation was the modular upregulation of a cobalamin-dependent gene cluster

**Table 6. Differential expression of the pyrimidine biosynthesis and salvage locus during co-cultivation of *L. monocytogenes* 6179 and cheese rind bacteria.**

| *L. monocytogenes* 6179 Locus_tag | *L. monocytogenes* EGD-e locus_tags | Gene | *L. monocytogenes* 6179 and *Psychrobacter* L7 12 h broth | *L. monocytogenes* 6179 and *Psychrobacter* L7 24 h plate | *L. monocytogenes* 6179 and *Brevibacterium* S111 12 h broth | *L. monocytogenes* 6179 and *Brevibacterium* S111 72 h plate |
|---|---|---|---|---|---|---|
| LM6179_2601 | lmo1831 | pyrE | 1.49 | 2.43 | 1.19 | ns* |
| LM6179_2602 | lmo1832 | pyrF | 1.75 | 2.75 | 1.29 | 2.90 |
| LM6179_2603 | lmo1833 | pyrD | 1.75 | 2.76 | 1.24 | 2.98 |
| LM6179_2604 | lmo1834 | pyrK | 1.58 | 2.27 | 1.05 | 2.74 |
| LM6179_2605 | lmo1835 | pyrAB | 1.94 | 2.54 | 1.32 | 2.72 |
| LM6179_2606 | lmo1836 | pyrAA | 1.88 | 2.12 | 1.24 | 2.92 |
| LM6179_2607 | lmo1837 | pyrC | 1.41 | 2.05 | 0.90 | 2.52 |
| LM6179_2608 | lmo1838 | pyrB | ns* | 2.12 | ns* | 2.43 |
| LM6179_2609 | lmo1839 | pyrP | 1.76 | 2.88 | ns* | 2.80 |
| LM6179_2610 | lmo1840 | pyrR | 1.47 | ns* | ns* | 2.18 |

*ns denotes that the gene was not DE in that condition

Values show the log2 fold changes in gene expression.

(CDGC, Table 7). Modules of the CDGC are involved in the catabolism of 1,2-propanediol (*pdu* genes, Fig 2), ethanolamine (*eut* genes, Fig 3), and the import and biosynthesis of cobalamin (*cbi/cob* genes Fig 4). Cobalamin is an essential cofactor of key *pdu* and *eut* catabolism genes [92, 93]. After 12 h co-cultivation in broth, *pdu*, *eut*, and *cbi/cob* genes of the CDGC were upregulated in response to both cheese bacteria. However, this occurred to a lesser degree during growth with *Brevibacterium* S111 (18 genes upregulated) than with *Psychrobacter* L7 (70 genes upregulated). Furthermore, co-cultivation of *L. monocytogenes* 6179 and *Psychrobacter* L7 for 24 h on plates exhibited the upregulation of several CDGC genes (n = 34) similar to the 12 h broth co-cultivations. After 72 h of co-cultivation with *Psychrobacter* L7 and *Brevibacterium* S111, several genes in the *pdu* module were significantly upregulated. In contrast to 12 h, none of the genes of the *eut* module were observed to be DE after 72 h. In *Salmonella enterica*, ethanolamine utilization is repressed by the 1,2-propanediol degradation genes, and *Salmonella* is hypothesized to prefer 1,2-propanediol catabolism over ethanolamine [94]. Possibly since ethanolamine degradation produces a more volatile toxic intermediate (acetaldehyde) than 1,2-propanediol catabolism, thereby leading to additional carbon loss and cellular toxicity [95, 96]. It is tempting to speculate that the upregulation of the *pdu* genes and not of the *eut* genes after 72 h of co-cultivation with both bacteria suggests that *L. monocytogenes*, like *Salmonella*, may prefer 1,2-propanediol catabolism, especially during exposure to presumably more severe stress conditions such as 72 h of co-cultivation in comparison to shorter periods. In general, expression of the *pdu* and *eut* CDGC modules are repressed when the cobalamin riboswitches Rli39 (*pdu*) and Rli55 (*eut*) fail to bind cobalamin and thus generate longer ncRNA transcripts. The long-form Rli39 transcript spans the antisense region of *pocR*, the positive regulator of the *pdu* genes. It thus represses PocR-mediated activation of the *pdu* genes by binding *pocR* mRNA [92]. The long-form transcript of Rli55 forms a secondary structure that sequesters the positive regulator of *eut* genes, EutV, near the 3' region of the ncRNA, thus hindering activation of the *eut* genes [93]. The Rli39 long-form was upregulated in co-cultivation with *Psychrobacter* L7 after 12 h, and the long-form of Rli55 was upregulated in both 12 h co-

**Table 7. Differential expression of the cobalamin-dependent gene cluster showing the number of upregulated DE genes during co-cultivation of *L. monocytogenes* 6179 with cheese rind bacteria.**

| | *pdu* genes: LM6179_1448 – LM6179_1477 | Rli39 long-form cobalamin riboswitch | *eut* genes: LM6179_1478 – LM6179_1494 | Rli55 long-form cobalamin riboswitch | *cob/cbi* genes: LM6179_1495 – LM6179_1516 | Rli57 putative cobalamin riboswitch |
|---|---|---|---|---|---|---|
| No. of predicted genes in CDGC module | 31 | | 18 | | 23 | |
| *L. monocytogenes* 6179 and *Psychrobacter* L7 12 h broth | 30 genes upregulated | gene upregulated | 18 genes upregulated | gene upregulated | 22 genes upregulated | gene upregulated |
| *L. monocytogenes* 6179 and *Psychrobacter* L7 24 h plate | Four genes upregulated | ns* | 16 genes upregulated | ns* | 14 genes upregulated | gene upregulated |
| *L. monocytogenes* 6179 and *Psychrobacter* L7 72 h plate | 10 genes upregulated | gene upregulated | ns* | ns* | 2 genes upregulated | ns* |
| *L. monocytogenes* 6179 and *Brevibacterium* S111 12 h broth | Three genes upregulated | ns* | Six genes upregulated | gene upregulated | Nine genes upregulated | gene upregulated |
| *L. monocytogenes* 6179 and *Brevibacterium* S111 72 h plate | 28 genes upregulated | gene upregulated | ns* | ns* | 15 genes upregulated | gene upregulated |

*ns denotes that the gene was not DE in that condition.

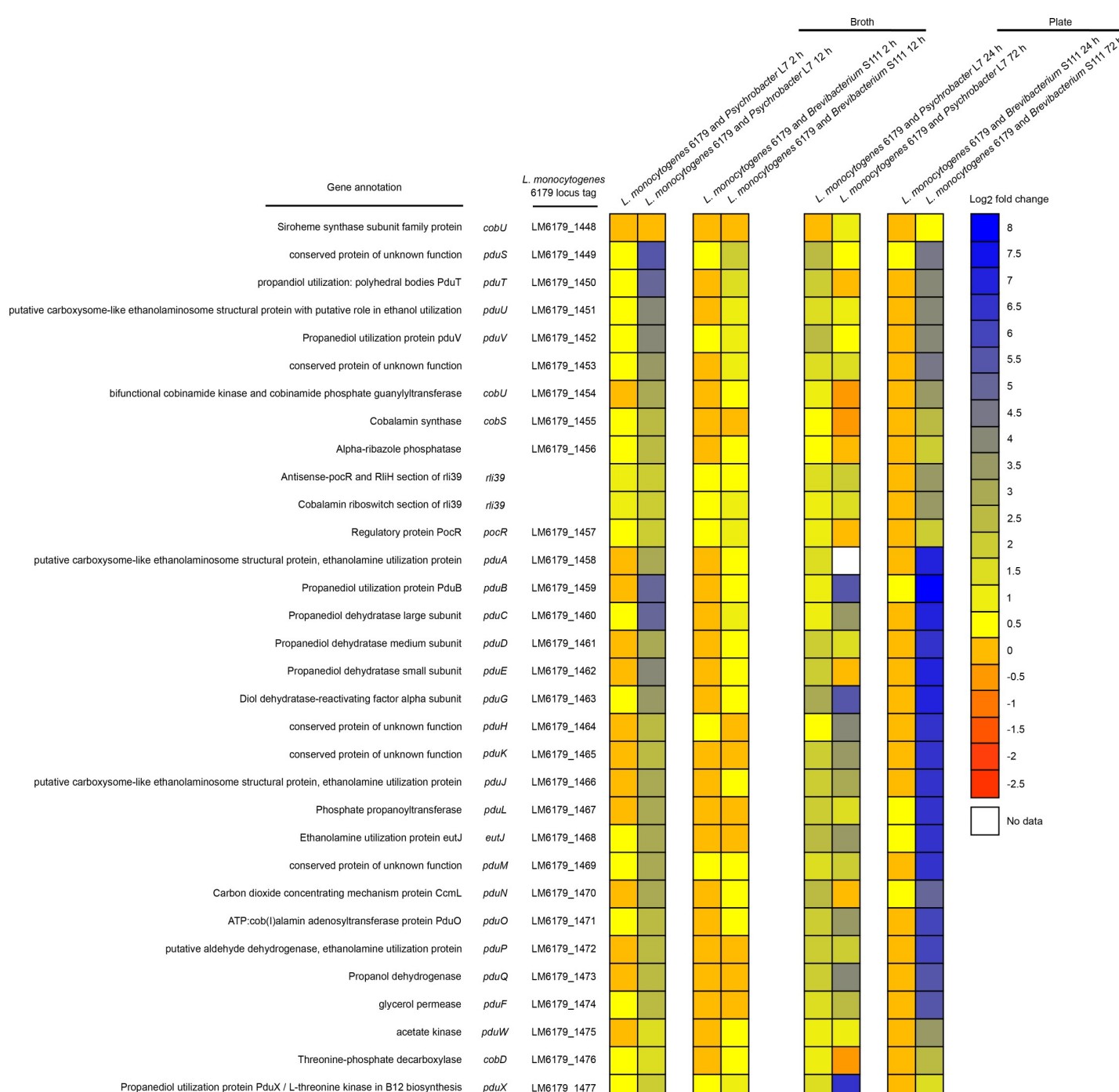

**Fig 2. Heatmap showing the gene expression of the *L. monocytogenes* pdu module, which is involved in the catabolism of 1,2-propanediol, during co-cultivation of *L. monocytogenes* 6179 with *Brevibacterium* S111 and *Psychrobacter* L7 in broth and on plates.** Colors represent the log2 fold change in gene expression corresponding to each co-culture condition compared to the monoculture control and the respective *pdu* module gene. "No data" indicates that no reads were mapped to the corresponding gene during the tested condition. Please see S1 and S2 Tables for more details regarding gene expression patterns, including q-values and log2 fold changes.

cultivations with *Psychrobacter* L7 and *Brevibacterium* S111. The simultaneous upregulation of the genes responsible for uptake and biosynthesis of cobalamin, and the upregulation of the

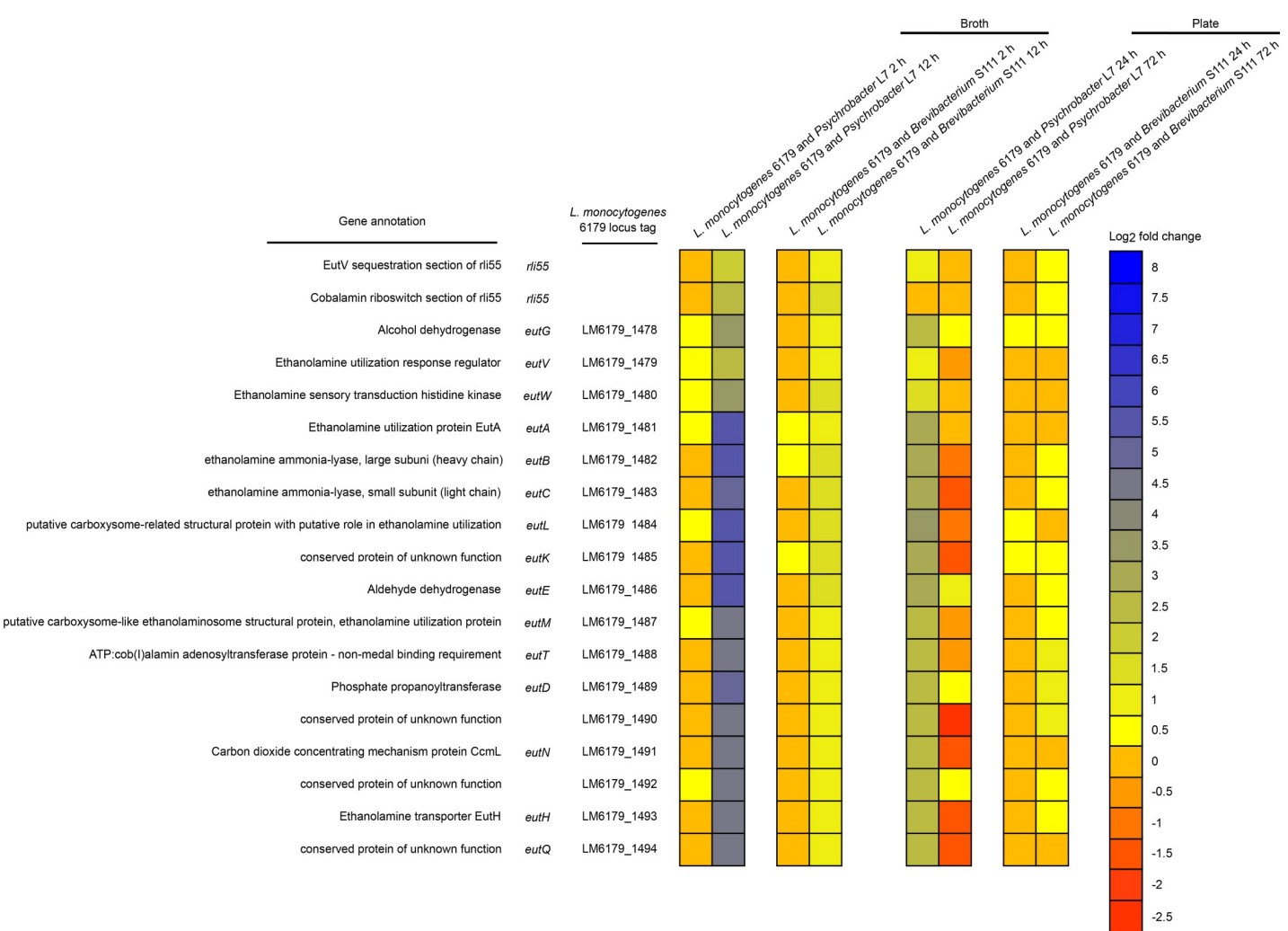

**Fig 3. Heatmap showing the gene expression of the *L. monocytogenes eut* module, which is involved in the catabolism of ethanolamine, during co-cultivation of *L. monocytogenes* 6179 with *Brevibacterium* S111 and *Psychrobacter* L7 in broth and on plates.** Colors represent the log2 fold change in gene expression corresponding to each co-culture condition compared to the monoculture control and the respective *eut* module gene. Please see S1 and S2 Tables for more details regarding gene expression patterns, including q-values and log2 fold changes.

*pdu* and *eut* gene clusters and their respective repressive long-form ncRNAs, may suggest the availability of propanediol and ethanolamine, and a developing shortage of free cobalamin after 12 h of co-culture. Notably, the long-forms of Rli39 and Rli55 were not upregulated after co-cultivation with *Psychrobacter* L7 for 24 h on plates, possibly indicating that more free cobalamin is available during this condition in comparison with the 12 h broth co-cultivations.

The CDGC is well conserved in *Listeria* species that colonize the gastrointestinal tract [97], and the *L. monocytogenes* CDGC and similar cobalamin-dependent gene clusters of other enteric pathogens are vital in pathogenicity [96, 98]. Additionally, the *L. monocytogenes* CDGC may have an essential role during exposure to food and food production relevant stress conditions [15, 30, 33, 99, 100]. Intriguingly, the CDGC is induced when *L. monocytogenes* is cultivated with other bacteria such as *Bacillus subtilis*, *Carnobacterium*, and *Lactobacillus* in environmental and mouse infection studies [15, 17, 18].

The present study is in line with previous research that indicates *L. monocytogenes* utilizes alternative substrates, such as ethanolamine and 1,2-propanediol, to increase its survival in

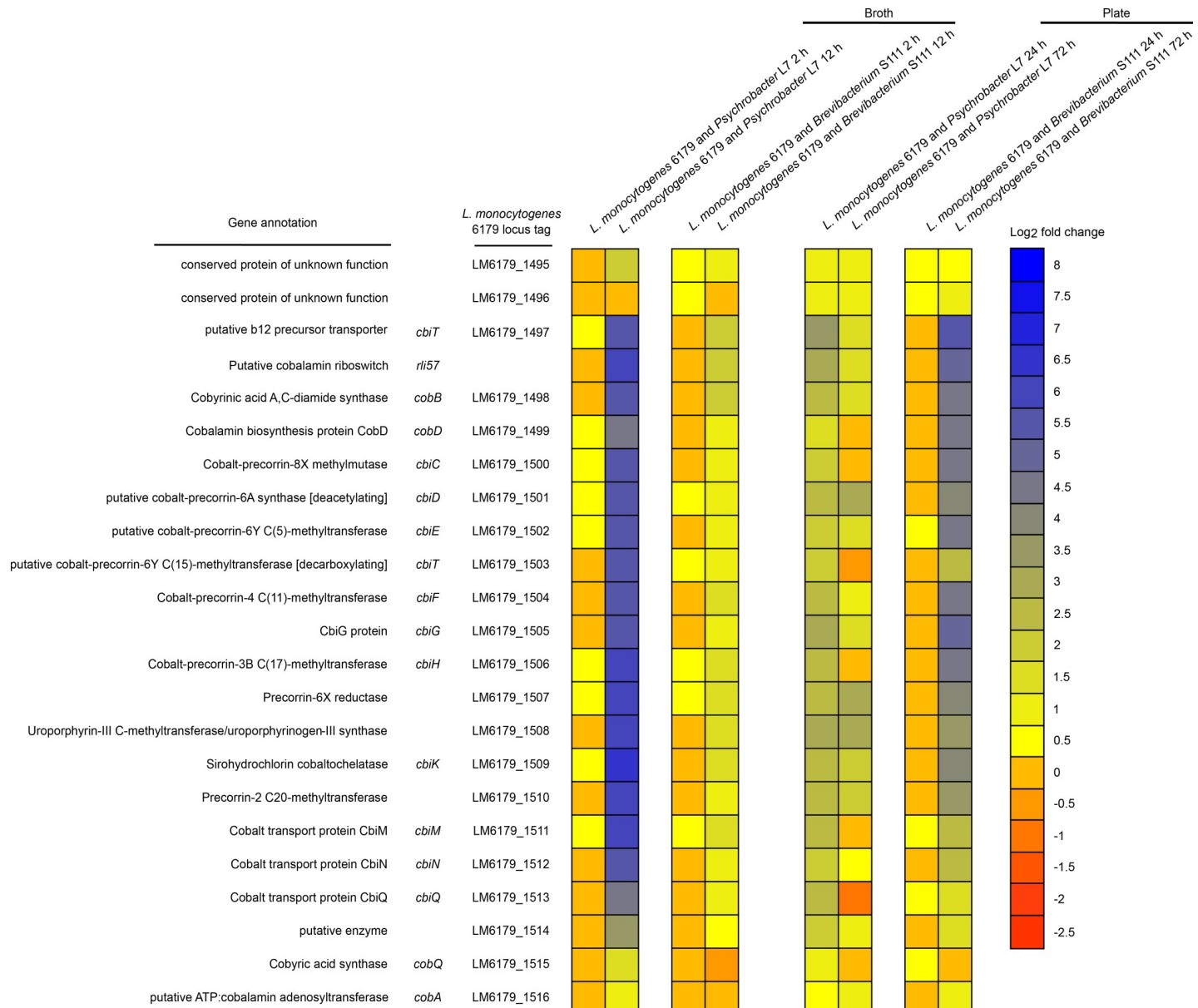

**Fig 4. Heatmap showing the gene expression of the *L. monocytogenes* *cob/cbi* module, which is involved in the production of cobalamin derivatives, during co-cultivation with *Brevibacterium* S111 and *Psychrobacter* L7 in broth and on plates.** Colors represent the log2 fold change in gene expression corresponding to each co-culture condition compared to the monoculture control and the respective *cob/cbi* module gene. Please see S1 and S2 Tables for more details regarding gene expression patterns, including q-values and log2 fold changes.

stressful and competitive niches [15, 17, 18, 91, 99]. Notably, the genes required for ethanolamine and 1,2-propanediol metabolism are absent from the genomes of *Psychrobacter* L7 and *Brevibacterium* S111. Thus, both strains cannot degrade ethanolamine and 1,2-propanediol, providing additional support for the importance of the *eut* and *pdu* genes for *L. monocytogenes* under these conditions. Recently, Marinho et al. [64] identified that the expression of six CDGC genes, specifically genes encoded by the *cob/cbi* and *pdu* modules, were influenced by Rli47. *pduX* (*LM6179_1477*, *lmo1170*) was the highest upregulated gene after co-cultivation of *L. monocytogenes* and *Psychrobacter* L7 for 72 h on plates (log2 fold increase of 6.47) and is a Rli47-dependent

gene. In *Salmonella*, PduX is an L-threonine kinase necessary for the *de novo* synthesis of a cobalamin derivative required for the utilization of 1,2-propanediol [96]. These data, combined with what is previously known of the CDGC, lead us to hypothesize that the CDGC may provide an advantage for *L. monocytogenes* in mixed cultures and, in part, is regulated by Rli47.

## Conclusion

This study raises the possibility that *L. monocytogenes* 6179 uses a multifaceted approach in adapting to co-culture conditions by inducing both aerobic and anaerobic metabolic pathways for energy acquisition. *Lap* was consistently among the most upregulated genes during co-culture, suggesting an increase in ethanol production. The ncRNA Rli47 was by far the highest expressed gene in this study. Other studies have shown that Rli47 influences several systems observed here to be induced by co-culture, suggesting an essential role of Rli47 in modulating metabolism in response to growth with other bacteria. Perhaps most notably, was the induction of the CDGC involved in the fermentation of ethanolamine and 1,2-propanediol. The utilization of ethanolamine and 1,2-propanediol may increase *L. monocytogenes* fitness when co-cultured with bacteria that are unable to metabolize them. Based on results from the scientific literature [64, 83–85, 89] and obtained in the current study, we have developed a scenario of pathways found to be consistently upregulated during coculture and where Rli47 may influence these systems (Fig 5). Further studies will be necessary to elucidate how the genes and

**Fig 5. Proposed model of differentially expressed metabolic systems of *L. monocytogenes* 6179 during co-cultivation with cheese rind bacteria and their relation to Rli47.** Bold indicates major metabolic and respiratory pathways. Gray dashed lines represent the hypothesized flow of formate, and blue paths show systems possibly regulated by Rli47 based on data from a number of recent studies [64, 83–85, 89]. Red dashed lines represent electron flow through proteins and the cellular membrane, and electrons are denoted by *e⁻*. Q represents membrane-bound menaquinone involved in electron shuttling.

pathways of interest identified in this study may contribute to *L. monocytogenes* fitness when exposed to other bacteria.

## Supporting information

**S1 Table. *Listeria monocytogenes* 6179 gene expression results after co-cultivation with *Psychrobacter* L7.**
(XLSX)

**S2 Table. *Listeria monocytogenes* 6179 gene expression results after co-cultivation with *Brevibacterium* S111.**
(XLSX)

**S3 Table. Prophage genes significantly upregulated after 72 h co-cultivation of *L. monocytogenes* 6179 on plates with *Psychrobacter* L7.**
(PDF)

**S4 Table. Selected differentially expressed genes showing consistent gene expression changes in multiple co-cultivation conditions.** Values show the log2 fold change of gene expression for the respective condition.
(PDF)

**S1 Fig. Principal component analyses to visualize variance between *L. monocytogenes* 6179 transcriptome replicates of broth co-cultivations and the respective monoculture controls.** Panels A. and B. correspond to broth co-cultivation replicates of *L. monocytogenes* 6179 and *Psychrobacter* L7 and their corresponding monoculture controls after 2 and 12 h incubation periods, respectively. Panels C. and D. correspond to broth co-cultivation replicates of *L. monocytogenes* 6179 and *Brevibacterium* S111 and their corresponding monoculture controls after 2 and 12 h incubation periods, respectively.
(PDF)

**S2 Fig. Principal component analyses to visualize variance between *L. monocytogenes* 6179 transcriptome replicates of plate co-cultivations and the respective monoculture controls.** Panels A. and B. correspond to plate co-cultivation replicates of *L. monocytogenes* 6179 and *Psychrobacter* L7 and their corresponding monoculture controls after 24 and 72 h incubation periods, respectively. Panels C. and D. correspond to plate co-cultivation replicates of *L. monocytogenes* 6179 and *Brevibacterium* S111 and their corresponding monoculture controls after 24 and 72 h incubation periods, respectively.
(PDF)

## Acknowledgments

We would like to extend our gratitude to Bienvenido Cortes for his extensive feedback during the development of this manuscript.

## Author Contributions

**Conceptualization:** Justin M. Anast, Stephan Schmitz-Esser.

**Data curation:** Justin M. Anast.

**Formal analysis:** Justin M. Anast, Stephan Schmitz-Esser.

**Funding acquisition:** Stephan Schmitz-Esser.

**Investigation:** Justin M. Anast, Stephan Schmitz-Esser.

**Methodology:** Justin M. Anast, Stephan Schmitz-Esser.

**Project administration:** Stephan Schmitz-Esser.

**Resources:** Stephan Schmitz-Esser.

**Supervision:** Stephan Schmitz-Esser.

**Validation:** Justin M. Anast.

**Visualization:** Justin M. Anast.

**Writing – original draft:** Justin M. Anast, Stephan Schmitz-Esser.

**Writing – review & editing:** Justin M. Anast, Stephan Schmitz-Esser.

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
