## [Decision Letter · Decision Letter 0]

16 Jun 2020

PONE-D-20-14512

The transcriptome of Listeria monocytogenes during co-cultivation with cheese rind bacteria suggests adaptation by induction of ethanolamine and 1,2-propanediol catabolism pathway genes

PLOS ONE

Dear Dr. Schmitz-Esser,

Thank you for submitting your manuscript to PLOS ONE. After careful consideration, we feel that it has merit but does not fully meet PLOS ONE’s publication criteria as it currently stands. Therefore, we invite you to submit a revised version of the manuscript that addresses the points raised during the review process.

We look forward to receiving your revised manuscript.

Kind regards,

Luca Cocolin

Academic Editor

PLOS ONE

Journal Requirements:

Reviewers' comments:

Reviewer's Responses to Questions

**Comments to the Author**

1. Is the manuscript technically sound, and do the data support the conclusions?

Reviewer #1: Yes

2. Has the statistical analysis been performed appropriately and rigorously? 

Reviewer #1: Yes

3. Have the authors made all data underlying the findings in their manuscript fully available?

Reviewer #1: Yes

4. Is the manuscript presented in an intelligible fashion and written in standard English?

Reviewer #1: Yes

5. Review Comments to the Author

Reviewer #1: The work deals with the study of the transcriptome of L. monocytogens during its co-cultivation with microbes isolated from cheese rind and how the pathogen is adapted under different conditions of co-cultivation. The manuscript is well written. The methodology is clear for repetition and the experiments were logically designed. The statistical analysis, interpretation and discussion of the results were correct. According to authors' statement all the data are available without restriction.

The only comment is for Table 1 in which the number of predicted plasmids is three while the number given for the predicted plasmid contig size(s) in kbp are four. Please explain with a footnote.

6. PLOS authors have the option to publish the peer review history of their article (what does this mean?). If published, this will include your full peer review and any attached files.

Reviewer #1: No

---

## [Author Response · Author response to Decision Letter 0]

26 Jun 2020

Reviewer #1: 

The work deals with the study of the transcriptome of L. monocytogens during its co-cultivation with microbes isolated from cheese rind and how the pathogen is adapted under different conditions of co-cultivation. The manuscript is well written. The methodology is clear for repetition and the experiments were logically designed. The statistical analysis, interpretation and discussion of the results were correct. According to authors' statement all the data are available without restriction.

The only comment is for Table 1 in which the number of predicted plasmids is three while the number given for the predicted plasmid contig size(s) in kbp are four. Please explain with a footnote.

Response: There are three predicted plasmids in the strain, but one of them is predicted to consist of two contigs. As suggested, we have clarified this in a footnote.

---

## [Editor Report · Decision Letter 1]

6 Jul 2020

The transcriptome of Listeria monocytogenes during co-cultivation with cheese rind bacteria suggests adaptation by induction of ethanolamine and 1,2-propanediol catabolism pathway genes

PONE-D-20-14512R1

Dear Dr. Schmitz-Esser,

We’re pleased to inform you that your manuscript has been judged scientifically suitable for publication and will be formally accepted for publication once it meets all outstanding technical requirements.

Kind regards,

Luca Cocolin

Academic Editor

PLOS ONE
---

## [Editor Report · Acceptance letter]

10 Jul 2020

PONE-D-20-14512R1 

The transcriptome of *Listeria monocytogenes* during co-cultivation with cheese rind bacteria suggests adaptation by induction of ethanolamine and 1,2-propanediol catabolism pathway genes 

Dear Dr. Schmitz-Esser:

I'm pleased to inform you that your manuscript has been deemed suitable for publication in PLOS ONE. Congratulations! Your manuscript is now with our production department. 

Kind regards, 

on behalf of

Dr. Luca Cocolin 

Academic Editor

PLOS ONE